# Common Buckthorn Engineered Biochar (Rhamnus Cathartica), Calcined Quagga Mussel Shells (Dreissena Rostriformis), Pickled Steel, and Steel Slag as Filter Media for the Sorption of Phosphorus from Agricultural Runoff

**Michael A. Holly [1,\*], Joseph R. Sanford [2], Patrick S. Forsythe [3], Marcia R. Silva [4], Daniel D. Lakich [4], Camryn K. Swan [3] and Keenan A. Leonard [3]**

[1] Richard J. Resch School of Engineering, College of Science, Engineering and Technology, University of Wisconsin-Green Bay, 2420 Nicolet Dr, Green Bay, WI 54311, USA
[2] School of Agriculture, College of Business, Industry, Life Science and Agriculture, University of Wisconsin-Platteville, 313 Pioneer Tower, 1 University Plaza, Platteville, WI 53818, USA
[3] Natural and Applied Sciences, College of Science, Engineering, and Technology, University of Wisconsin-Green Bay, 2420 Nicolet Dr, Green Bay, WI 54311, USA
[4] Water Technology Accelerator, Office of Research—Global Water Center, 247 W. Freshwater Way, Suite 717, Milwaukee, WI 53204, USA
\* Correspondence: hollym@uwgb.edu; Tel.: +1-(920)-465-2503

**Abstract:** The reuse of waste residuals as reactive media is a sustainable solution to remove phosphorus (P) from wastewater and reduce eutrophication. Large reactive waste media filters incorporated in edge-of-field treatment systems could reduce P loading from agricultural fields. We measured the treatment potential of regionally available waste residuals (i.e., calcined quagga mussel shells (CSHELL), magnesium activated biochar (MGBC), pickled steel (PSTEEL), and steel slag (SLAG)) for dissolved P removal. CSHELL and MGBC had elevated sorption capacities (64,419 and 50,642 mg kg$^{-1}$, respectively) in comparison to SLAG and PSTEEL (14,541 mg kg$^{-1}$ and 736 mg kg$^{-1}$, respectively). However, CSHELL requires long reaction times for removal (22% removal after 1.5 h) and P sorbed to MGBC is removed with DI, reducing treatment potential. SLAG and PSTEEL were the only media with significant reductions of agricultural runoff and had the greatest overall treatment potential. SLAG is recommended for removal and replacement systems while PSTEEL is suitable for larger systems designed for regeneration on site.

**Keywords:** filter media; phosphorus removal; agricultural runoff; constructed wetlands; wastewater reactive media; stormwater runoff

## 1. Introduction

Seasonal hypoxia in the Great Lakes (as a result of excessive algal production) has a detrimental effect on the local aquatic species, environment, and recreation. Despite investments in point reductions, seasonal hypoxia is persistent [1] due to agricultural sources of phosphorus (P) (estimated to account for 35% of total annual P load to Lake Michigan [2]). Notably, edge-of-field treatments (e.g., sedimentation basins, constructed wetlands, and buffer strips) have the potential to effectively trap nutrients from agricultural runoff year-round and prevent those nutrients from reaching receiving waters [3–5]. Sedimentation basins are effective in reducing coarse particles (>75 μm) [6]; however, these do not reduce dissolved P (DP) and additional treatment is required to reduce the remaining suspended solids and dissolved P (DP) in effluent [7]. Adsorption of P onto reactive media has treatment potential of DP as it requires a low installation cost, has high performance, and has minimal operation requirements [8].

Reactive media can reduce DP; however, a barrier to incorporating filter media for treatment of agricultural runoff is their susceptibility to clogging over time [9]. Media

clogging can be mitigated by pretreating runoff using sedimentation basins, which results in an agricultural runoff treatment system capable of removing particulate and dissolved P. Numerous media (>80) have been investigated for wastewater P removal and high P sorption potential of media corresponds with elemental content of Ca, Fe, Mg and Al [10]. The primary mechanism for P removal for previously successful media is precipitation by calcium. Industrial by-products have the highest measured P removal capacities (up to 420 g P kg$^{-1}$ for furnace slag), followed by natural materials (40 g P kg$^{-1}$ for heated opoka) [10]. The ideal reactive media for a phosphorus removal structure should have the following critical characteristics for a high rate of adoption: high P sorption capacity, adequate hydraulic conductivity, mechanical strength, fast P sorption kinetics, reusability, local availability, and low cost of production [11].

Reactive media created from waste products can effectively remove pollutants from wastewater [12,13], promote a circular economy [14], and could enable pervasive adoption for treatment systems. Common Buckthorn (*Rhamnus cathartica*), quagga mussels (*Dreissena bugensis*) shells, steel slag (SLAG), and steel turnings are locally available in the Midwest and were selected as raw materials for the study. Buckthorn, mussel shells, and steel turnings are unreactive and require thermal or chemical treatment for P sorption. Biochar is an effective and economical media for the removal of P and sorption is dependent on metal cations available on the surface of biochar [15]. Previously studied lanthanum-modified activated carbon from pine cones had a measured sorption capacity of 68.2 mg g$^{-1}$ [16]. However, rare earth activation of biomass would be costly and could hinder deployment of biochar as reactive media; therefore, less costly metals should be evaluated. Thermal treatment (calcification) of mussel shells increases sorption capacity, as white clam shells heated to 800 °C resulted in a potential substrate for P removal (max sorption capacity 38.7 mg g$^{-1}$) [11]. High temperature (>700 °C) pyrolysis is energy intensive and low temperature calcification of shells would be beneficial to adoption. A pretreatment for steel known as pickling is a common practice to remove scale from steel surfaces prior to phosphating steel for painting [17]; pickled steel turnings may result in a mechanically strong reactive media with P removal potential. Currently, limited information exists on the treatment potential of the modification techniques (i.e., low temperature thermal treatment of shells, pretreatment of steel through pickling) and feedstocks (i.e., buckthorn wood waste) to create reactive media for the removal of P from agricultural runoff. The objective of the present study is to measure the applicability of modified waste residues (magnesium activated biochar [MGBC], calcined quagga mussel shell [CSHELL], pickled steel turnings [PSTEEL], and [SLAG]) as reactive media for P removal through laboratory adsorption measurements.

## 2. Materials and Methods

### 2.1. Materials

Media used for evaluation were ground and/or sieved to obtain a particle size between 2 and 4 mm for adequate hydraulic conductivity. Common Buckthorn and quagga mussels, an invasive species to Northeast Wisconsin, were collected from local parks and beaches near the University of Wisconsin–Green Bay. Common Buckthorn was dosed with magnesium chloride and thermally treated to create an engineered biochar (MGBC) using procedures modified from Yin et al. [18]. Collected Buckthorn was mulched, soaked in magnesium chloride hexahydrate (MgCl$_2$·6H$_2$O) (190 g L$^{-1}$, at a 1:5 residual to solution ratio) for 6 h, dried for 4 h (60 °C), and heated at 350 °C for one hour in a reactor purged with nitrogen gas. A method similar to Ngyuen et al.'s [11] was used to treat mussel shells. In the current study, mussel shells were rinsed with DI water, dried for 24 h at 60 °C, and heated to 350 °C for one hour in a reactor purged with nitrogen gas to produce CSHELL [11]. Steel turnings were collected from a local welding company out of Green Bay, Wisconsin, and prepared using a method developed from the pickling and descaling process [17]. Steel turnings were heated in a (41 °C) HCl and DI water solution (100 g HCl L$^{-1}$, at a 1:2 residual to solution ratio) for 15 min and, then rinsed with DI water and dried for

4 h (60 °C). SLAG was obtained from an electric arc furnace producing steel billets in the Midwest.

*2.2. Experimental Methods*

2.2.1. Characterization of Materials

Raw and treated media were tested to analyze changes in physical or chemical properties due to treatment methods. Surface area, pore size, and pore volumes for materials were calculated with the Brunauer–Emmett–Teller (BET) method using a Quantachrome gas sorption analyzer (Quantachrome Instruments, Boynton Beach, FL, USA). Samples were degassed at 200 °C for 330 min and then tested with nitrogen adsorption at 770 °K. Multiple tests were performed using Fourier Transform Infrared Spectroscopy (FTIR) to determine material characteristics. Samples were ground into fine powders before being tested on the IRTracer-100 Shimadzu single reflection ATR accessory (Shimadzu Corporation, Kyoto, Japan). The surfaces of the reactive media were observed using a scanning electron microscope (Hitachi S-4800 field emission scanning electron microscope [Hitachi Global, Tokyo, Japan]). The samples were fixed on aluminum stubs using adhesive tape. All the samples were scanned under high vacuum conditions at an accelerating voltage of 10 kV (500× magnification).

2.2.2. Batch Testing

Adsorption tests were completed in vails with working standards of P and an adsorbent dose of 1.5 g of media to 30 mL of solution. Vials were shaken (220 rpm) with an orbital shaker and then filtered through A-645 filters. The experimental time for all trials was 24 h, excluding the kinetic trial. Experiments were completed in triplicate including controls (working standard only). Phosphorus concentrations were analyzed in controls and filtrate using discrete analyzer (AQ 300, Seal Analytical, Southampton, United Kingdom) according to U.S. EPA method 365.1.

Isotherm Study

Adsorption isotherm characteristics were quantified through incremental concentrations of P (2.5, 5, 10, 50, 100, 200, 400, 500, 800, and 1000 mg L$^{-1}$). Working standards were created from a stock P solution prepared by dissolving $KH_2PO_4$ in DI water and adjusting the pH of the solution to 8 using NaOH. Data analysis included calculating media sorption (Equation (1)) and fitting data to Langmuir and Freundlich models (Equations (2) and (3)):

$$S = \frac{(C_i - C_{eq})}{m} \qquad (1)$$

$$S = \frac{S_{max} K_L C_{eq}}{1 + K C_{eq}} \qquad (2)$$

$$S = K_f C_{eq}^{\frac{1}{n}} \qquad (3)$$

where $S$ (mg P g$^{-1}$) is calculated sorption, $C_i$ and $C_{eq}$ (mg P L$^{-1}$) are the initial and final (equilibrium) concentrations, $m$ is the mass of media, $S_{max}$ (mg P g$^{-1}$) is the estimated sorption capacity, $K_L$ (L mg$^{-1}$) is the Langmuir sorption binding strength coefficient, $K_f$ (mg$^{1-1/n}$ L$^{1/n}$ g$^{-1}$) is the Freundlich sorption constant, and $n$ (dimensionless) is the intensity of sorption. The Langmuir and Freundlich models are non-linear, and fitting the data to the models requires iterative solutions [19]. Measured data was fitted to the Langmuir and Freundlich models using a Microsoft Excel (2016) spreadsheet developed by Bolster and Hornberger [19]. The spreadsheet calculates best fit parameters, standard errors, and goodness of fit measures.

Kinetic Study

The reaction rate of P sorption was determined for a low and high strength wastewater concentration (2 mg P $L^{-1}$ and 50 mg P $L^{-1}$, respectively). Working standards were created similarly to the isotherm study using $KH_2PO_4$ in DI water and adjusting to a pH of 8. Vials were shaken for different adsorption times from 0.1 to 24 h and immediately filtered through A-645 filters. Pseudo first- and second-order kinetic models were used to fit the measured data through (Equations (4) and (5), respectively):

$$ln(S_e - S_t) = \ln(S_e) - k_1 t \tag{4}$$

$$\frac{t}{S_t} = \frac{1}{k_2 S_e^2} + \frac{1}{S_e} t \tag{5}$$

where $S_e$ and $S_t$ are the calculated sorption at equilibrium (24 h) and at each experimental time, respectively, $k_1$ ($h^{-1}$) and $k_2$ ($(mg/g)^{-1}/h$) are the adsorption rates of the first and second rates, respectively, and $t$ is the time (h). Additional models used to fit data included the Intra-particle diffusion, Boyd, and Elovich models (not shown for each media). The Weber and Morris model was used for intra-particle diffusion analysis (Equation (6)):

$$q_t = K_p \sqrt{t} + C \tag{6}$$

where $K_p$ is a rate constant (mg $kg^{-1}$ $min^{-0.5}$) and $C$ is boundary layer thickness.

pH and Real Agricultural Runoff Performance Study

The effect of pH on P adsorption was determined similarly to isotherm trials by varying the initial pH of the 50 mg P $L^{-1}$ solution between 4 and 12 using HCl and NaOH at various concentrations. Examination of efficacy in real wastewater was completed using edge-of-field runoff from a corn and soy field receiving dairy manure in Northeastern Wisconsin in place of a working standard solution. An additional set of runoff samples were spiked (50 mg $L^{-1}$) to evaluate the treatment potential for high strength P wastewater.

Desorption Study

Desorption efficiency was completed on dried (4 h at 60 °C) recovered media from the 50 mg $L^{-1}$ and 100 mg $L^{-1}$ vials from the isotherm trial. To determine the strength of sequestration media, important to leachability and recovery, media desorption trials included subsequent solutions of removal strength. Initially, media received 30 mL of DI and were shaken for 24 h (220 rpm), filtered, and analyzed for P concentration. The experiment was then repeated on the recovered media receiving DI with a 0.5 M sodium bicarbonate ($NaHCO_3$, to represent plant available P) and then 1 M potassium hydroxide (KOH) extraction solution.

*2.3. Statistical Analysis and Evaluating Applicability of Filter Medias*

Statistical analysis was completed using SAS 9.4 (SAS Institute, Cary, NC, USA). Adsorption test triplicates for each treatment were evaluated to assess the statistical difference using a one-way analysis of variance and Tukey's honest significance test (HSD). A method by Nguyen et al. [11] was used to evaluate the overall treatment potential of media based on critical characteristics [11]. Criteria were weighted according to their significance. Each media were ranked as high = 3, medium = 2, or low = 1, and a cumulative score for media evaluated was calculated [11].

### 3. Results

*3.1. Characterization of Materials*

MGBC and CSHELL had a larger surface area (7.9, 5.8 $m^2$ $g^{-1}$, respectively) than SLAG and PSTEEL (3.2 and 0.3 $m^2$ $g^{-1}$, respectively) (Table 1). SEM images indicate that some of the outer surface of the raw Buckthorn (Figure 1a) was removed after treatment into biochar

(Figure 1b). However, the pyrolysis was not high enough to expose the biological capillary structure, as previously seen for higher temperatures >550 °C [20]. Characterization of mussel shell surface indicates multiple layers, which is expected for this sample. Grain boundaries are observed on the surface of PSTEEL (Figure 1e) when compared to raw steel (Figure 1d), as expected due to the pickling process. The thermal conversion of Buckthorn increases the surface area eightfold (surface area of untreated Buckthorn chips 1.0 m$^2$ g$^{-1}$), increasing treatment potential. FTIR results illustrate combustions' impact on the functional groups of the Buckthorn chips and the conversion of CaCO$_3$ to CaO for the mussel shells (Figure 2).

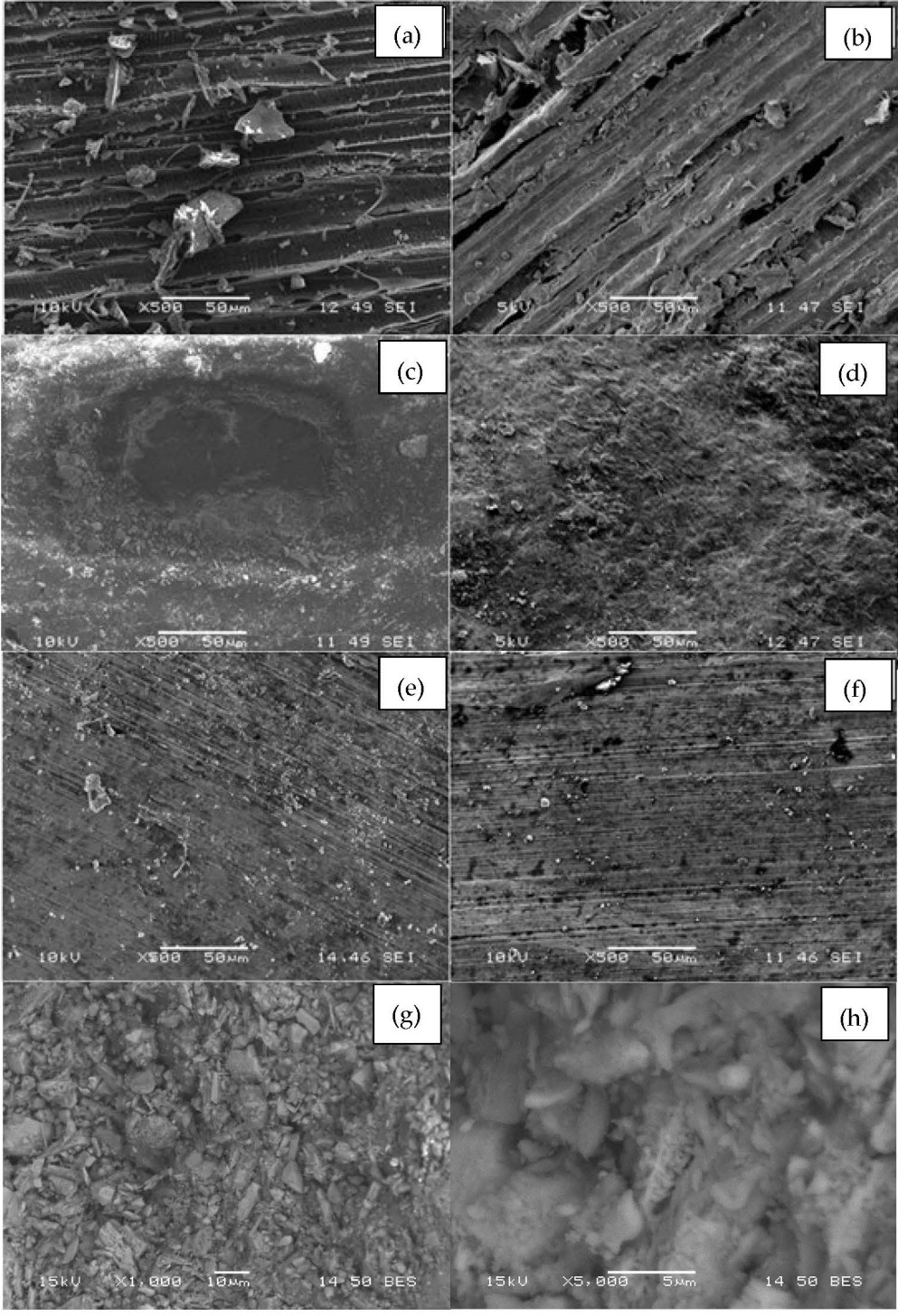

**Figure 1.** SEM images of each material's surface taken at 500× magnification of: (**a**) raw buckthorn; (**b**) MGBC, (**c**) raw mussel shell; (**d**) CSHELL; (**e**) raw steel; (**f**) PSTEEL; (**g,h**) SLAG.

**Table 1.** Pore size, pore volume, and surface area of raw and treated reactive media.

| Media | Pore Size (Radius Å) | Pore Volume (cc g$^{-1}$) | Surface Area (m$^2$ g$^{-1}$) |
|---|---|---|---|
| Raw buckthorn | 15.6 | 0.0050 | 1.0 |
| MGBC | 17.5 | 0.0210 | 7.9 |
| Raw shell | 19.5 | 0.0110 | 6.0 |
| CSHELL | 19.7 | 0.0140 | 5.8 |
| Raw steel | 17.5 | 0.0000 | 0.1 |
| PSTEEL | 16.5 | 0.0000 | 0.3 |
| SLAG | 16.2 | 0.0150 | 3.2 |

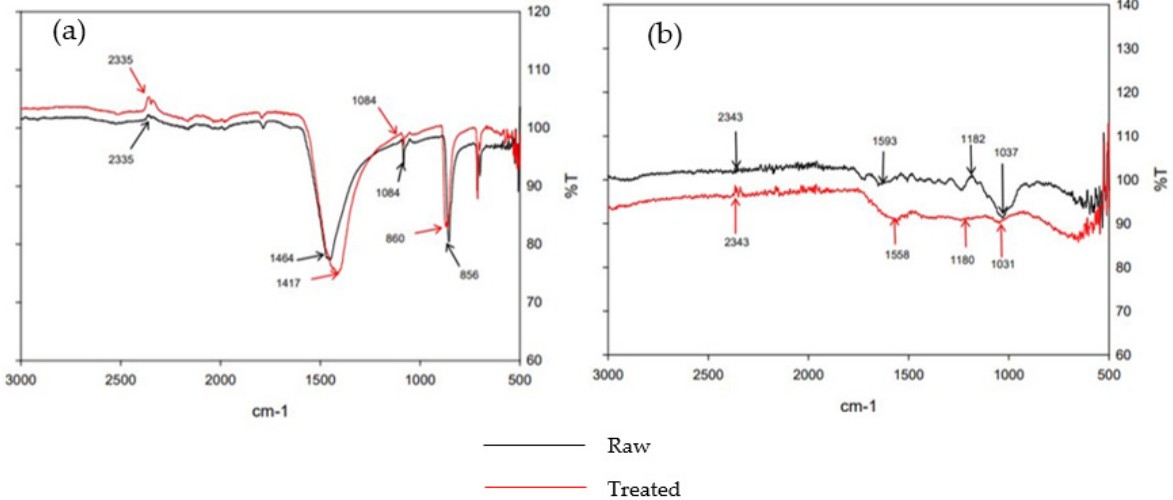

**Figure 2.** FTIR graphs showing the difference in raw and treated samples: (**a**) CSHELL; (**b**) MGBC.

*3.2. Adsorption Isotherm Study*

Sorption capacities for media ranged from 736 to 64,419 mg kg$^{-1}$, Table 2. CSHELL had significantly greater sorption capacity (64,419 mg kg$^{-1}$) than SLAG (14,541 mg kg$^{-1}$) and PSTEEL (736 mg kg$^{-1}$) (Figure 3, Table 1). The sorption capacity for MGBC (50,642 mg kg$^{-1}$) was statistically similar to CSHELL and PSTEEL. Notably, the P sorption by CSHELL was variable at higher P capacities (>2000 mg P L$^{-1}$).

**Table 2.** Adsorption isotherm and kinetic characteristics of reactive media. Treatments with similar letters (a and b) did not show statistical significance at the $\alpha = 0.05$ level.

| Isotherm Constants and Coefficients of Determination | | | | | | |
|---|---|---|---|---|---|---|
| Isotherm model | Langmuir | | | Freundlich | | |
| Media | $q_m$ (mg kg$^{-1}$) | $K_L$ (L mg$^{-1}$) | $R^2$ | $K_F$ (mg kg)(L kg$^{-1}$)$^{1/n}$ | $1\,n^{-1}$ | $R^2$ |
| CSHELL | 64,419 [a] | 0.0005 | 0.84 | 91 | 1.18 | 0.92 |
| MGBC | 50,642 [ab] | 0.0019 | 0.92 | 2561 | 3.11 | 0.96 |
| SLAG | 14,541 [bc] | 0.003 | 0.86 | 602 | 2.36 | 0.87 |
| PSTEEL | 736 [c] | 0.0269 | 0.84 | 70 | 2.58 | 0.86 |
| Kinetic constants and coefficients of determination for the 50 mg P L$^{-1}$ trials | | | | | | |
| Kinetic model | Pseudo first-order model | | | Pseudo second-order model | | |
| Media | $q_e$ (mg kg$^{-1}$) | $K_1$ (1 h$^{-1}$) | $R^2$ | $q_e$ (mg kg$^{-1}$) | $k_2$ (g mg$^{-1}$ h$^{-1}$) | $R^2$ |
| CSHELL | 0.15 | 0.67 | 0.67 | 0.16 | 24 | 0.99 |
| MGBC | 232 | 0.04 | 0.04 | 1250 | 0.01 | 0.99 |
| SLAG | 1.09 | 0.39 | 0.90 | 0.78 | 8.93 | 0.92 |
| PSTEEL | 0.11 | 0.92 | 0.84 | 0.2 | 0.49 | 0.37 |

**Table 2.** *Cont.*

| Isotherm Constants and Coefficients of Determination | | | | | |
|---|---|---|---|---|---|
| Kinetic constants and coefficients of determination for the 2 mg P L$^{-1}$ trials | | | | | |
| Kinetic model | Pseudo first-order model | | | Pseudo second-order model | | |
| Media | $q_e$ (mg kg$^{-1}$) | $K_1$ (1 h$^{-1}$) | $R^2$ | $q_e$ (mg kg$^{-1}$) | $k_2$ (g mg$^{-1}$ h$^{-1}$) | $R^2$ |
| CSHELL | 0.05 | 0.00 | 0.00 | 0.11 | 160 | 0.98 |
| SLAG | 0.004 | −1.33 | 0.38 | 0.03 | 350 | 0.99 |
| PSTEEL | 0.03 | 0.01 | 0.25 | 0.27 | 7.14 | 0.39 |

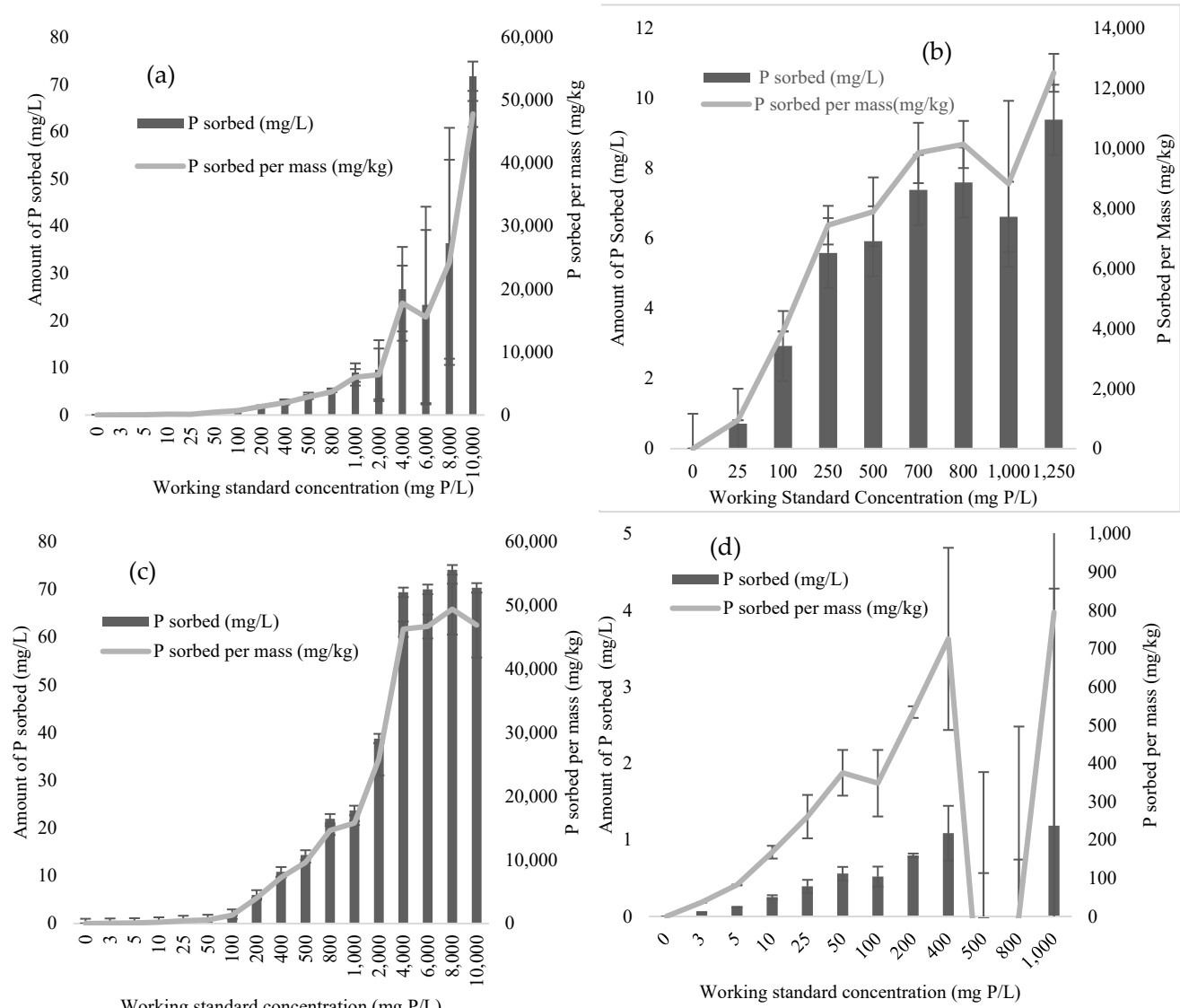

**Figure 3.** Sorption with increasing concentration for: (**a**) CSHELL; (**b**) SLAG; (**c**) MGBC; (**d**) STEEL.

### 3.3. Impact of pH

Sorption of P was dependent on pH for CSHEEL and PSTEEL (Figure 4A). CSHELL removal at a pH of 10 was significantly lower than removal for solutions at a pH of 4, 6, and 12. The highest removal for CSHELL occurred at a pH of 4, which showed significantly greater removal of P than the other pH solutions tested. PSTEEL had significantly greater removal at a pH of 4 and 6 than other pH solutions tested with minimal removal (10%) at a pH of 12. SLAG was unaffected by pH (4 to 12) with >95% removal of 50 mg P/L.

MGBC removal at a pH of 12 was significantly lower than the other pH solutions; however, it maintained P sorption potential (93% removal of P).

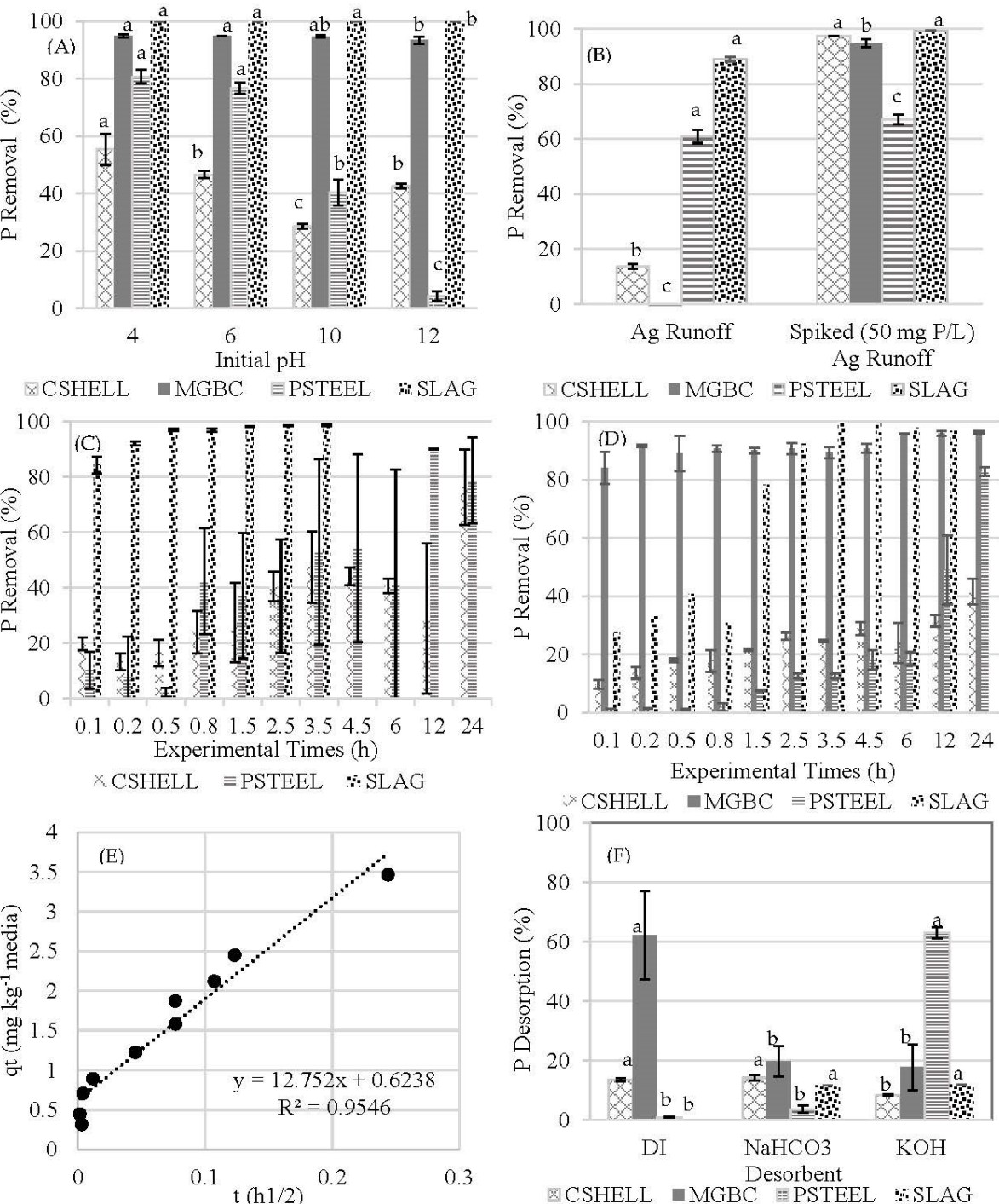

**Figure 4.** Batch sorption testing: (**A**) impact of pH; (**B**) agricultural runoff; (**C**) low P (2 mg P/L) concentration kinetics; (**D**) high concentration (50 mg P/L) kinetics; (**E**) intraparticle diffusion modeling of high concentration PSTEEL trial; (**F**) sequential desorption (DI, sodium bicarbonate, and potassium hydroxide) of media collected from 50 mg P L$^{-1}$ batch isotherm trials. P removal between pH values for individual treatments with similar letters (a, b, and c) did not show statistical significance at the $\alpha = 0.05$ level. Desorption between extractants for individual treatments with similar letters (a, and b) did not show statistical significance at the $\alpha = 0.05$ level. P removal from agricultural runoff between treatments with similar letters (a, b, and c) did not show statistical significance at the $\alpha = 0.05$ level.

### 3.4. Reaction Rate

MGBC removed P rapidly in the high concentration kinetics trail with an 84% removal efficiency of P in the first six minutes and 96% after 24 h (Figure 4D). However, MGBC had low to no removal at low concentrations of P. SLAG had rapid P removal efficiency in the low concentration trial, with 85% of P removed in the first six minutes. CSHELL and PSTEEL required greater contact time for high removal efficiencies to be reached, with removal of just 22% and 8% after 1.5 h in the high concentration trials and 42% and 83% removal after 24 h, respectively.

Kinetics of an adsorption reaction can be fit to linear models to elucidate on the sorption mechanisms of a reaction. For both the low and high concentration kinetic trials, CSHELL, MGBC, and SLAG were best fit by the pseudo second-order models (Table 2), indicating chemisorption was the dominant removal mechanism. Kinetics for PSTEEL were best fit by the intraparticle diffusion model (Figure 3E), and the two linear segments in the graph suggest multiple controlling mechanisms. The first mechanism as seen for the initial linear segment can be attributed to boundary diffusion which occurred during the first 30 min of the reaction. The second linear section represents gradual chemisorption between phosphorus and iron of the steel.

### 3.5. Desorption

DI water removed 61% of P sorbed to MGBC, significantly more than P desorbed by the $NaHCO_3$ (20%) and KOH (18%) solutions (Figure 4F), indicating weak sorption of P. P sorbed to PSTEEL was removed primarily by KOH (63%) significantly more than by solutions of DI and $NaHCO_3$. SLAG had minimal P removed from the DI solution, and removal by $NaHCO_3$ and KOH solutions (11%) was statistically similar. Significantly more P was desorbed from the CSHELL by solutions of DI and $NaHCO_3$ (13 and 14%, respectively) than the KOH solution (8%). SLAG had the lowest desorption of sorbed P (24%) from the three desorbent solutions followed by CSHELL (36%). Desorption of P from PSTEEL (68%) and MGBC (100%) was more complete.

### 3.6. Efficacy of the Media in Agricultural Runoff

SLAG and PSTEEL had the greatest removal of P from the 24 h agricultural runoff batch trials (89 and 61% P removal from agricultural runoff, respectively), and significantly greater removal than CSHELL (14%) and MGBC (<0%) (Figure 2B). Phosphorus was released from the biochar (>2.1 mg P $L^{-1}$), which negated treatment of agricultural runoff (0.4 mg P $L^{-1}$). SLAG, CSHELL, and MGBC demonstrated the greatest treatment potential for high strength P wastewater [99, 97 and 95% removal from spiked agricultural runoff (77 mg P $L^{-1}$), respectively], which was significantly greater than PSTEEL (67%).

## 4. Discussion

CSHELL had the greatest sorption capacity of media tested (64,419 mg $kg^{-1}$) and was comparable to other previously evaluated material with elevated sorption of P, including naturally occurring materials (300 to 51,000 mg $kg^{-1}$), industrial byproducts (830 to 107,000 mg $kg^{-1}$), and man-made materials (2500 to 12,000 mg $kg^{-1}$) [11]. Adsorption capacity for CSHELL was larger than previously evaluated calcined white clam mussel shells heated to 800 °C [11], with a previously measured sorption capacity of 38,700 mg $kg^{-1}$. A higher sorption capacity would be beneficial to deployment, by increasing time to media exhaustion, and is recommended for systems where less frequent removal or regeneration of material is desired. Notably, the sorption to the mussel shell was variable at higher P capacities (>2000 mg P $L^{-1}$) and may have been due to variability in the calcification of the material. CSHELL was slightly less effective at a neutral pH (including agricultural runoff); however, it is effective for both acidic and basic wastewater, as similarly found by Nguyen et al. [11]. A couple different forms of calcium phosphate minerals form at a pH between 7 and 9, including precursors to the thermodynamically stable calcium hydroxyapatite, in comparison to direct formation of calcium hydroxyapatite at a pH > 9 [21]. Predominantly,

the use of CSHELL as a reactive media is negatively impacted by a low reaction rate and limited removal of P at low concentrations. CSHELL filters would have to be larger, in comparison to the other evaluated media, to accommodate an extended reaction time. Effluent from CSHELL filters would also require additional treatment for a more complete removal of P at lower concentrations. Although not effective as a filter media, CSHELL could be an effective reactive media for in situ immobilization of P from wastewater storages with elevated concentrations of P (e.g., livestock wastewater or surface waters with elevated dissolved P).

Adsorption capacity for MGBC was lower than Al modified biochar's (aluminum modified biochar, sorption capacity of 74,470 mg kg$^{-1}$) [18]; however, the sorption capacity was similar to other Mg modified biochar (56,000 mg kg$^{-1}$ [22]). Magnesium biochar was unaffected by pH and, therefore, would be applicable to runoff from fields across soil types. Advantages of MGBC also included a short reaction time, which would reduce the footprint of reactive media treatment systems. Notably, P sorbed to MGBC was weakly bound, as the majority of P sorbed was removed with DI water, negating treatment potential. Mg-biochar from coffee ground waste demonstrated a strong bond with a measured bioavailability of 9 to 43% [22]. Higher desorption from the mg-biochar from the current study created from wood could be a result of a lower pore volume (0.021 vs. 0.11 cc g$^{-1}$), surface area (7.9 vs. 36.4 m$^2$ g$^{-1}$), and average pore size (17.5 vs. 116.5 Å) for the wood biochar compared to the coffee bean biochar. The porosity of the coffee beans may reduce the desorption of phosphate bound to mg-biochar by limiting exposure to desorbent solutions. Biochar as reactive media should be created under conditions that maximize porosity (e.g., pyrolysis > 400 °C) to reduce unfavorable desorption of P from reactive media. MGBC was unable to remove P at low concentrations (0.4 mg P/L) as seen for agricultural runoff and from the isotherm (no removal below 0.9 mg P L$^{-1}$); therefore, it is not suitable as reactive media for low concentrations of P. Suspended solids or other ions released form the biochar may be inhibiting removal at low concentrations.

Adsorption capacity for PSTEEL was the lowest amongst the studied media (736 mg kg$^{-1}$). However, the pickling process is still recommended for steel filter media as Si oxides densely form on the steel during the manufacturing process, significantly affecting the sorption of P [23]. Iron filings are successful as reactive media for urban runoff with an adsorption capacity of 4800 mg kg$^{-1}$ and it is likely that the adsorption capacity of steel turnings will increase with oxidation. Steel reactive media will rust as it is exposed to oxygen and water in the field; therefore, removal may increase over time. To the authors' knowledge, the current study is the first attempt at using steel as a reactive media for P removal. PSTEEL had a lower reaction rate; therefore, it requires longer contact times and larger media systems, similar to CSHELL. However, PSTEEL was capable of removing P at low concentrations and is capable of removing P from agricultural runoff. Future studies should evaluate the treatment potential of steel turnings over time to understand sorption capacity with oxidation.

SLAG used for the current study had a measured sorption capacity (14,541 mg kg$^{-1}$) similar to previously studied electric arc furnace SLAG (10,480 mg kg$^{-1}$ [24]). Removal of P by SLAG is highly variable, as this material is unambiguously defined and will vary in chemical composition. SLAG has been successful in removing P at the field scale [24–26], as confirmed by the measured short reaction time, removal of P at low concentrations, including in agricultural runoff, and a stronger bond to removed phosphorus. Notably, less than 25% of P was removed by desorption solutions, and SLAG that reaches exhaustion in the field will have to be excavated in lieu of regeneration in the field [26]. Field deployment of SLAG is also impacted by clogging, as bicarbonate in tile drainage will react with calcium in SLAG, precipitating calcium carbonate, which can fill pore space [26].

Media were evaluated for their agricultural runoff treatment potential through weighted criteria, including agricultural runoff treatment efficacy, sorption capacity, mechanical strength, P removal kinetics, reuse potential, availability, energy consumptions, and costs. All evaluated material had disadvantages and scored lower (18 to 31) than the score for

the ideal media (36) (Table 3). SLAG had the highest potential for adoption of the media analyzed; however, it is limited by sorption capacity and reusability. Although SLAG is not reusable, it is of a low cost and can be beneficially used as an aggregate after exhaustion. Due to a low reusability, SLAG should only be used in systems designed for removal and replacement of media. However, due to a fast reaction, a SLAG treatment system would only require a minimum of 6 min to treat wastewater with a phosphorus concentration of 1 mg/L. PSTEEL demonstrated limited potential as a media (27), limited by sorption capacity, kinetics, and cost. PSTEEL would require a larger treatment system; however, it could be regenerated in situ. CSHELL and MGBC are not recommended as filter media (22 and 18, respectively), as these presented a low efficacy for treatment of agricultural runoff.

**Table 3.** Filter media screening scores for ideal and evaluated media.

| Evaluation Criteria | Weight | Ideal Media | | CSHELL | | EZEO | | MGBC | | PSTEEL | |
|---|---|---|---|---|---|---|---|---|---|---|---|
| | | Rank | Score | Rank | Score | Rank | Score | Rank | Score | Rank | Score |
| Efficacy in ag runoff | 3 | High | 9 | Low | 3 | High | 9 | None | 0 | High | 6 |
| P Sorption capacity | 2 | High | 6 | High | 6 | Low | 2 | High | 6 | Low | 2 |
| Mechanical strength | 2 | High | 6 | Low | 2 | High | 6 | Low | 2 | High | 6 |
| P removal kinetics | 1 | High | 3 | Medium | 2 | High | 3 | High | 3 | Low | 1 |
| Reuse potential | 1 | High | 3 | Medium | 2 | Low | 1 | Low | 1 | High | 3 |
| Availability in Midwest | 1 | High | 3 | Medium | 2 | Low | 1 | High | 3 | High | 3 |
| Energy consumption | 1 | Low | 3 | Medium | 2 | High | 1 | High | 1 | Low | 3 |
| Cost | 1 | Low | 3 | Low | 3 | High | 1 | Medium | 2 | High | 1 |
| CUMULATIVE SCORE | | | 36 | | 22 | | 24 | | 18 | | 25 |

For all criteria except energy consumption and cost: High = 3, Medium = 2, Low = 1, and None = 0; For the energy consumption and cost media: High = 1, Medium = 2, and Low = 1.

## 5. Conclusions

Modified waste residuals (mussel shell, wood chips, steel turnings, and SLAG) were evaluated for P removal as reactive media for agricultural runoff treatment. The waste materials selected were from locally available low-costing waste materials capable of promoting a circular economy. Of the materials, SLAG and PSTEEL are recommended for deployment in agricultural runoff systems although both have notable disadvantages. SLAG is recommended for smaller treatment systems with a minimum hydraulic retention time of 6 min, capable of removal and replacement. PSTEEL requires larger treatment systems with a minimum hydraulic time of 12 h for 80% removal; however, it could be regenerated on site. Treatment systems utilizing a combination of media could improve the performance of waste residuals and should be evaluated in future experiments. Notably, recommendations are limited by the experimental design parameters (batch adsorption, P concentration, and pH) of the study; therefore, SLAG and PSTEEL should be evaluated at the field scale to verify potential.

**Author Contributions:** Conceptualization, M.A.H. and J.R.S.; methodology, M.A.H.; software, M.A.H.; validation, M.A.H.; formal analysis, M.A.H.; investigation, M.A.H., J.R.S., P.S.F., D.D.L., C.K.S. and K.A.L.; resources, M.A.H.; data curation, J.R.S., D.D.L., C.K.S. and K.A.L.; writing—original draft preparation, M.A.H., M.R.S., D.D.L., C.K.S. and K.A.L.; writing—review and editing, M.A.H., P.S.F. and J.R.S.; visualization, M.A.H.; supervision, M.A.H., P.S.F. and J.R.S.; project administration, M.R.S. and M.A.H.; funding acquisition, M.R.S. and M.A.H. All authors have read and agreed to the published version of the manuscript.

**Funding:** This research was funded by the Freshwater Collaborative of Wisconsin: Track 4: Freshwater Research Experience for Undergraduate Awards.

**Data Availability Statement:** Not applicable.

**Acknowledgments:** Adsorption measurements were completed by the spring 2021 University of Wisconsin–Green Bay environmental science capstone course. We would like to acknowledge Ryan Curtice, Annissa Derbique, Miranda Esser, Carly Flunker, Emily Hearley, Lese McWey, Emma Loucks, Richard Perschon, Grace Steele, Mary Stewart, Emily Walter, Hannah Wentzel, and Malory Zickert for their assistance in measurements and preliminary data analysis. A special thanks to Steven Hardcastle at the University of Wisconsin Milwaukee's Advanced Analysis Facility for physical and chemical analysis of media. Finally, we would also like to acknowledge Bonnie Allen, Technical Writer, for her peer review and editing of the document.

**Conflicts of Interest:** The authors declare no conflict of interest.

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
