# Peer review of "Common Buckthorn Engineered Biochar (Rhamnus Cathartica), Calcined Quagga Mussel Shells (Dreissena Rostriformis), Pickled Steel, and Steel Slag as Filter Media for the Sorption of Phosphorus from Agricultural Runoff"

_conservation, doi:10.3390/conservation2040047_

Round 1

Reviewer 1 Report

Please mention reference to the method you followed in processing the materials (Line 77-92).

The flow needs to be improved.

Author Response

Please mention reference to the method you followed in processing the materials (Line 77-92).

AU: Complete, additional references to methods used were added and text was clarified.

The flow needs to be improved.

AU: The manuscript has been peer reviewed and edited for readability.

Reviewer 2 Report

In this paper, the authors developed waste-derived materials for phosphorus sorption. Overall, the experimental results are interesting and would attract great scientific interest in related fields. The paper is well presented, and the results are convincing. Before acceptance, the following issues should be addressed.
1. The introduction is not well-written, and some important issues are ignored.

(1)   Page 1, current status of P recovery and removal should be discussed, especially related methods, here are some suggested ref. (Water research, 2022, 209: 117891.; Journal of Cleaner Production, 2022: 133439.; Water, 2021, 13(4): 517.; Science of The Total Environment, 2022: 153750.; Science of the Total Environment, 2021, 786: 147437.)

(2)   Also, the advantage of sorption over other methods should be discussed (Water Research, 2022, 221: 118820.; Environmental Pollution, 2021, 280: 116995.; Journal of Cleaner Production, 2022: 133676.; Eco-Environment & Health, 2022, 1, 86-104.; Carbohydrate Polymers, 2021, 274: 118671.)

(3)   Page 2, lines 57-58, where it discusses ways of waste reutilization and circular economy; this part is not robust. This has to be substantially improved with most-up-to date literature as from some recent studies (e.g., Resources, Conservation and Recycling, 2022, 178: 106037.; Applied Catalysis B: Environmental, 2021, 298: 120583.; Green Chemistry, 2021,23, 6538-6547; Journal of Hazardous Materials, 2022, 431: 128590.; Environmental Functional Materials, 2022, 1: 34-48.).

2. Page 2, line 83: what does 1:5 g mL-1 mean?

3. Page 3, line 114, why KH2PO4 was used as the P source?

4. Page 4, line 174, the discussion does not match the picture, check carefully.

5. Page 5, why DI water has different desorption effects on different materials? Give a short discussion.

6. As suggested here, the four materials perform differently in terms of sorption capacity and reusability, etc. A combination of some materials may help to improve the overall performance. The authors can briefly discuss this issue in the paper.

7. There are some grammatical errors in the paper, especially the used units. Please improve the quality of English and highlight all changes that you have made in the revised version for improving readability. 

Author Response

Thank you for the opportunity to revise and resubmit our manuscript.  We found the review to be helpful and we have made edits to address the concerns. Below, in italics, are our responses to the specific comments.

  1. The introduction is not well-written, and some important issues are ignored.
  • Page 1, current status of P recovery and removal should be discussed, especially related methods, here are some suggested ref. (Water research, 2022, 209: 117891.; Journal of Cleaner Production, 2022: 133439.; Water, 2021, 13(4): 517.; Science of The Total Environment, 2022: 153750.; Science of the Total Environment, 2021, 786: 147437.)

AU: The authors appreciate the additional references on P recovery and have added suggested references applicable to edge of field treatment of agricultural runoff.

  • Also, the advantage of sorption over other methods should be discussed (Water Research, 2022, 221: 118820.; Environmental Pollution, 2021, 280: 116995.; Journal of Cleaner Production, 2022: 133676.; Eco-Environment & Health, 2022, 1, 86-104.; Carbohydrate Polymers, 2021, 274: 118671.)

AU: The authors have added a statement in support of adsorption for treatment of agricultural runoff using the suggested references.

  • Page 2, lines 57-58, where it discusses ways of waste reutilization and circular economy; this part is not robust. This has to be substantially improved with most-up-to date literature as from some recent studies (e.g., Resources, Conservation and Recycling, 2022, 178: 106037.; Applied Catalysis B: Environmental, 2021, 298: 120583.; Green Chemistry, 2021,23, 6538-6547; Journal of Hazardous Materials, 2022, 431: 128590.; Environmental Functional Materials, 2022, 1: 34-48.).

AU: Additional support added to the statement on waste utilization and circular economy.

  1. Page 2, line 83: what does 1:5 g mL-1mean?

AU: 1 to 5 residual to solution ratio, the authors added additional text to clarify.

  1. Page 3, line 114, why KH2PO4was used as the P source?

AU: This is the salt form of phosphate which is dissolved in DI water to create PO4.

  1. Page 4, line 174, the discussion does not match the picture, check carefully.

AU: The authors agree and have revised the discussion.

  1. Page 5, why DI water has different desorption effects on different materials? Give a short discussion.

AU: Discussion on DI removal for MGBC is presented in the discussion section. A short discussion statement was added to results.

  1. As suggested here, the four materials perform differently in terms of sorption capacity and reusability, etc. A combination of some materials may help to improve the overall performance. The authors can briefly discuss this issue in the paper.

 AU: The authors agree and appreciate the reviewer’s comment and added a statement to the discussion. “Treatment systems utilizing a combination of medias could improve the performance of waste residuals and should be evaluated in future experiments.”

  1. There are some grammatical errors in the paper, especially the used units. Please improve the quality of English and highlight all changes that you have made in the revised version for improving readability. 

 AU: The authors have revised the manuscript to improve grammar and correct units.

Reviewer 3 Report

Acceptable

Author Response

Thank you for your review. The manuscript has been peer reviewed and edited.

Reviewer 4 Report

This work investigated the performance of different materials such as biochar (Rhamnus cathartica), calcined quagga mussel shells (Dreissena rostriformis), pickled steel and steel slag phosphoric acid-activated carbonaceous materials to remove phosphorus from agricultural runoff. The authors did much work. However, some issues need to be minor revised before considering acceptance.

1.     What was the pH of the solution at equilibrium for the different residual wastes studied?

2.     Is the stable material in an acidic environment?

3.     Can the materials be recycled multiple times?

4.     The novelty of this research should be well presented in the Introduction. Several relative papers are suggested to be cited:  (https://doi.org/10.3390/separations9100283; https://doi.org/10.1002/pat.5676; https://doi.org/10.3390/app11115125).

5.     It is suggested that the author briefly explain this work's enlightenment to other researchers.

6.     In terms of form and layout: Distribute your text evenly across the margins

Author Response

Thank you for the opportunity to revise and resubmit our manuscript.  We found the review to be helpful and we have made edits to address the concerns. Below, in italics, are our responses to the specific comments.

This work investigated the performance of different materials such as biochar (Rhamnus cathartica), calcined quagga mussel shells (Dreissena rostriformis), pickled steel and steel slag phosphoric acid-activated carbonaceous materials to remove phosphorus from agricultural runoff. The authors did much work. However, some issues need to be minor revised before considering acceptance.

  1. What was the pH of the solution at equilibrium for the different residual wastes studied?

AU: Thank you for your question. The solution was adjusted to a pH of 8 using NaOH as specified in the methods

  1. Is the stable material in an acidic environment?

AU: The authors understand the importance of pH for the batch testing. The pH was adjusted to a pH of 8 representative of the pH of agricultural runoff.

  1. Can the materials be recycled multiple times?

AU: The authors evaluated the desorption of P from the material; however, the reuse of desorbed material was not included in the current study. Desorption is the first step for reuse and was variable between media and the reuse potential was included in the discussion for each material

“Notably, less than 25% of P was removed by desorption solutions and slag that reaches exhaustion in the field will have to be excavated in lieu of regeneration in the field.”

  1. The novelty of this research should be well presented in the Introduction. Several relative papers are suggested to be cited:  (https://doi.org/10.3390/separations9100283; https://doi.org/10.1002/pat.5676; https://doi.org/10.3390/app11115125).

AU: The authors added an additional statement to the introduction to highlight the novelty of the research.

“Currently, limited information exists on the treatment potential of reactive media created from the presented modification techniques (i.e. low temperature thermal treatment of shells, pretreatment of steel through pickling) and feedstocks (i.e. buckthorn wood waste) for the removal of P from agricultural runoff.”

  1. It is suggested that the author briefly explain this work's enlightenment to other researchers.

AU: The authors added an additional statement to the conclusion to support enlightenment from the study

  1. In terms of form and layout: Distribute your text evenly across the margins

AU: The authors formatted tables to distribute text across margins.

Reviewer 5 Report

The paper is an exciting use of different organic and inorganic materials to remove P from agricultural run-off. However, there are several problems that make the document hard to understand

1. The title is hard to read; authors must re-write it to be more precise

Materials and methods.

2. Authors must divide into small subsections, which will improve the readability of the methods employed (i.e., 2.1. organisms; 2.2 physical and chemical properties, and others)

Results.

3. Authors should use the same subsections as in materials and methods

4. There are two "figure 3."

Discussion

5. the discussion presented by the authors could be more profound than all the analyses. The authors must rewrite this section since the same problem remains in the last two sections. There is too much information that is not well organized.

Author Response

Thank you for the opportunity to revise and resubmit our manuscript.  We found the review to be helpful and we have made edits to address the concerns. Below, in italics, are our responses to the specific comments.

The paper is an exciting use of different organic and inorganic materials to remove P from agricultural run-off. However, there are several problems that make the document hard to understand

  1. The title is hard to read; authors must re-write it to be more precise

AU: The authors revised the manuscript title.

Materials and methods.

  1. Authors must divide into small subsections, which will improve the readability of the methods employed (i.e., 2.1. organisms; 2.2 physical and chemical properties, and others)

AU: Complete.

Results.

  1. Authors should use the same subsections as in materials and methods

AU: Complete.

  1. There are two "figure 3."

AU: Thank you for the edit. This has been corrected.

Discussion

  1. the discussion presented by the authors could be more profound than all the analyses. The authors must rewrite this section since the same problem remains in the last two sections. There is too much information that is not well organized.

AU: Thank you for your comment. The authors organized the discussion by media. The authors believe the current discussion is adequate and within the limitations from the experimental design parameters (batch adsorption, P concentration, and pH). Flow through testing and field testing are needed to verify the treatment potential of reactive media evaluated in the study. Results are best representative of agricultural runoff treatment and additional trials are needed to evaluate treatment potential of other wastewater streams.

Round 2

Reviewer 2 Report

Accept.

Reviewer 5 Report

Authors have improved the quality of the document